# Effect of Thermal Conductive Fillers on the Flame Retardancy, Thermal Conductivity, and Thermal Behavior of Flame-Retardant and Thermal Conductive Polyamide 6

**DOI:** 10.3390/ma12244114

**Published:** 2019-12-09

**Authors:** Fang Wang, Wenbo Shi, Yuliang Mai, Bing Liao

**Affiliations:** Guangdong ProvincialKey Laboratory of Industrial Surfactant, Guangdong Research Institute of Petrochemical and Fine Chemical Engineering, Guangzhou 510665, China; fiona.wang@gdcri.com (F.W.); shiwenbo@jinbai-tpe.com (W.S.);

**Keywords:** polyamide 6, flame retardancy, thermal conductivity, thermal conductive fillers, composites

## Abstract

In this work, polyamide 6 (PA6) composites with improved flame retardancy and thermal conductivity were prepared with different thermal conductive fillers (TC fillers) such as aluminum nitride (AlN) and boron nitride (BN) in a PA6 matrix with aluminum diethylphosphinate (AlPi) as a fire retardant. The resultant halogen-free flame retardant (HFFR) and thermal conductive (TC) PA6 (HFFR-TC-PA6) were investigated in detail with a mechanical property test, a limiting oxygen index (LOI), the vertical burning test (UL-94), a cone calorimeter, a thermal gravimetric analysis (TGA) and differential scanning calorimetry (DSC). The morphology of the impact fracture surface and char residue of the composites were analyzed by scanning electron microscopy (SEM). It was found that the thermal conductivity of the HFFR-TC-PA6 composite increased with the amount of TC fillers. The TC fillers exerted a positive effect for flame retardant PA6. For example, the HFFR-TC-PA6 composites with the thickness of 1.6 mm successfully passed the UL-94 V-0 rating with an LOI of more than 29% when the loading amount of AlN-550RFS, BN-SW08 and BN-NW04 was 30 wt%. The morphological structures of the char residues revealed that TC fillers formed a highly integrated char layer surface (without holes) during the combustion process, as compared to that of flame retardant PA6/AlPi composites. In addition, the thermal stability and crystallization behavior of the composites were studied.

## 1. Introduction

Modified engineering materials have been widely used for electronic devices, automobiles and aerospace applications in the past few decades [1,2,3,4]. However, the cutting-edge compact, complex and integrated designs of devices have raised the requirements for engineering materials which have to bear the ability to be assembled in a small integrated space, light weight, super heat dissipation, and flame retardancy properties [5,6]. Even when metal and ceramic materials fit well in heat dissipation and flame retardancy properties, their extreme processing conditions and high densities lead them to be disqualified for such applications [7]. Polymer composites (e.g., polyamide 6 (PA6)), on the other hand, which have many excellent properties, such as an ease of processing, lightweight, good electrical insulation and tailorable mechanical properties [8,9], have been considered as the most promising candidates for compact and integrated electronics and new energy device applications, even if challenges regarding their thermal conductivity and fire retardancy still need to be addressed.

Polyamide 6 (PA6) is an important engineering plastic that has been vastly used due to its excellent mechanical properties, good heat resistance, and good chemical resistance [10,11,12,13,14,15,16]. PA6 can be easily applied to automobile components, electrical connectors, switch components, high power light emitting diodes (LEDs) field, etc. [17]. However, its applications are restricted to those fields that require high flame retardance and thermal conductivity due to its inadequate flame retardancy, severe flammable melt dripping, and relative low thermal conductivity [18,19]. Taking its application in electrical products as an example, during the usage of electrical products, the high temperature generated from circuits or other components is required to be emitted instantly, otherwise overheating may occur, thus sacrificing performance. Therefore, it is necessary to ensure that the PA6 used for such applications has good thermal conductivity. Additionally, the electrical components using PA6 are required to have high flame retardancy to avoid fires during electrical breakdown or other abnormal conditions. With these considerations, exploring the PA6 composites with both flame retardancy and thermal conductivity is of academic significance and practical value. However, thus far, most research has focused on either the fire retardancy or thermal conductive properties of PA6; limited reports have combined these two properties together in single studies, which are of course significant for applications. Yuan D. et al. [20] investigated the thermal conductivity of a PA6/PA66 1:1 blend by aluminum nitride (AlN) and found thermal conductivity of the PA6/PA66/AlN composite with 50% AlN was 1.5 W/mK, but the flame retardancy of the composite materials was not discussed. Zhong Y. et al. [21] reported the effect of boron nitride (BN) on the flame retardancy and thermal stability of flame retardant PA6, but the thermal conductivity of the flame-retardant-thermal conductive-PA6 (FR-TC-PA6) composites was not fully discussed.

To date, varieties of halogen-free flame retardants have been proposed for polyamide, including ammonium polyphosphate (APP) [22], melamine cyanurate (MCA) [23,24], and aluminum diethylphosphinate (AlPi) [25,26]. Among them, AlPi is phosphorus flame retardant which has been developed and commercialized. This material has been proven to be an effective flame retardant for PA6 [27,28]. Wirasaputra A. et al. [25] reported that the PA6/AlPi composites passed the UL-94 (vertical burning test) V-2 rating, and the limiting oxygen index (LOI) value reached up to 29.3% when 13 wt% AlPi was loaded. Ma K. et al. [19] reported that when the loading of AlPi was 13 wt%, thee LOI value of the PA6/AlPi composites increased to 30% and passed the UL-94 V-0 rating. However, these reports only focused on the flame retardancy properties of PA6; the thermal conductivity of flame retardant PA6 was not discussed.

A simple and effective method to enhance the thermal conductivity of a polymer matri, is to directly incorporate thermally conductive (TC) fillers, such as aluminum nitride (AlN) [20], boron nitride (BN) [29], alumina (Al_2_O_3_) [30], silicon carbide (SiC) [31], carbon nanotubes [32], and graphene [33]. Research has been conducted to evaluate the properties of the polymer/TC composites, and strategies have been proposed to improve the thermal conductivity of the polymer/TC composites by improving the dispersion of TC fillers in a polymer matrix. For example, Nikoo G. et al. [34] used the relationship between normalized storage modulus and angular frequency as a method to compare the degree of dispersion of BN particles in a PA6 and cyclic olefin copolymer (COC) matrix; they found that BN particles had a better dispersion in the PA6 matrix than in the COC matrix due to better compatibility between PA6 and BN. Tang D. et al. [35] studied the dispersion of KH550-modified BN in an epoxy matrix and its effect on the thermal conductivity and mechanical properties of the composite; it was found that KH550-BN could be more uniformly dispersed in the matrix than pristine BN, and the epoxy/KH550-BN composite had a relatively higher thermal conductivity. In addition, different processing methods have been found to have different effects on the thermal conductivity of composites. It has been found that the fillers in the melt are oriented by the flow during extrusion stretching or injection molding, and this leads to different thermal conductivities for anisotropic fillers [36].

As inorganic fillers, TC fillers may have either favorable or unfavorable effect on the flame retardancy of polymer composites, but reports on related research are limited. Almeras X. et al. [37] investigated the effect of fillers (talc and calcium carbonate) on the fire performance of a polypropylene/ammonium polyphosphate/polyamide-6 blend. It was found that the fire performance strongly depended on the nature of the fillers used, as talc increased and calcium carbonate decreased the flame retardancy performance of the corresponding composites.

In this study, we prepared halogen-free flame retardant (HFFR)-TC-PA6 composites bearing both a high thermal conductivity and excellent flame retardancy for potential application in an electrical field. The HFFR-TC-PA6 composites with fire retardant AlPi and different types of TC fillers were compounded by using extrusion and injection molding, and the effect of TC fillers on the mechanical properties, thermal conductivity, flame retardance properties and thermal behavior of the HFFR-TC-PA6 composites were investigated.

## 2. Experimental

### 2.1. Materials

Commercial grade PA6 pellets (1013B) were sourced from Ube Industries, Ltd., Tokyo, Japan. The flame retardant aluminum diethylphosphinate (AlPi) Exolit OP1230 was purchased from Clariant Co., Frankfurt, Germany. Thermal conductive filler (TC filler) aluminum nitride (two grades AlN, a spherical AlN-550RFS with an average particle size of 55 µm and aspherical AlN-300SFS with an average particle size of 10 µm) and boron nitride (two grades BN, a BN-NW04 with an average particle size of 0.5 µm and BN-SW08 with an average particle size of 6 µm) were purchased from Global Top Trading Co., Ltd. (Kunshan, China).

### 2.2. Preparation of HFFR-TC-PA6 Composites

The composites of HFFR-TC-PA6 were prepared by melt mixing through a twin-screw co-rotating extruder (model AK36, L/D ratio = 40, Nanjing Keya Chemical Complete Equipment Co., Ltd, Nanjing, China). The mixing ratios are given in Table 1. Firstly, PA6 was dried under vacuum at 100 °C for 24 h, while AlPi, AlN-550RFS, AlN-300SFS, BN-SW08 and BN-NW04 were all dried at 80 °C for 24 h before use. Secondly, the dried PA6, AlPi, and the TC fillers were mixed in a high-speed mixer at a speed of 1200 rpm for 2 min, the mixtures were fed into twin-screw extruder, the temperature profiles were set to 220, 225, 230, 240, 240, 240, 240, 240, 240 and 240 °C, from hopper to die, and then the extrudates were pelletized after cooling. After drying at 110 °C for 24h, the HFFR-TC-PA6 pellets were injection-molded to standard testing bars by using an injection molding machine (model TW-25V, clamping force 25 tons, Dongguan Taiwang Machinery Co., Ltd. Dongguan, China). The temperature profiles were set to 220 °C (hopper), 230 °C and 240 °C (nozzle).

### 2.3. Characterization

Mechanical properties were evaluated by tensile, flexural and Izod impact tests. Tensile and flexural tests were performed on an MTS-CMT 6104 universal mechanical tester (MTS, Eden Prairie, MN, USA). The dimension of the type 1A dumbbell-shaped specimens for tensile test was 150 × 20 × 4.0 mm in accordance with ISO 527: 2019, the speed of the test was 50 mm/min, the dimensions of rectangular bars for the flexural test were 80 × 10 × 4.0 mm in accordance with ISO 178: 2016, and the speed of the test was 2 mm/min. The notched Izod impact test was carried out with the MTS-ZBC 8400-B impact tester (MTS, Eden Prairie, MN, USA), the dimensions of the specimens with a type A notch were 80 × 10 × 4.0 mm in accordance with ISO 180: 2016. Each value of the mechanical properties was an average of five specimens.

The vertical burning test was performed according to UL-94, and the dimensions of all specimens were 125 × 13 × 3.0 mm and 125 × 13 × 1.6 mm. The limiting oxygen index (LOI) values were measured according to ISO 4589-2: 2017, and the dimensions of all samples were 80 × 10 × 4.0 mm. According to ISO 5660-1: 2015, the fire behavior of the samples was measured by a cone calorimeter device. Samples having size of 100 × 100 × 3.0 mm were exposed to a radiant cone (50 kW/m^2^). The plaques were placed in the sample holder with a retainer frame, and the top surface of the sample was directly exposed the heat source.

The thermal conductivity of the HFFR-TC-PA6 composite was measured using the steady state heat flow method and conducted on a Longwin instrument (model LW-9389, Longwin company, Dongguan, China). Measurements were performed according to ASTM D5470-17, and the dimensions of the samples were 27 × 27 × 5 mm. This method is based on idealized heat conduction between two parallel, isothermal surfaces separated by a test specimen of uniform thickness. The thermal gradient imposed on the specimen by the temperature difference between the two contacting surfaces causes the heat flow through the specimen. This heat flow is perpendicular to the test surface and is uniform across the surfaces with no lateral heat spreading. The thermal conductivity was calculated using the following equation:(1)$$\lambda =\frac{Q_{h}+Q_{c}}{2}\cdot \frac{L}{\Delta T}$$
where *λ* is the thermal conductivity of the material, *Q_h_* is the heat flow in hot meter bar, *Q_c_* is the heat flow in cold meter bar, *L* is the thickness of the sample, and Δ*T* is the temperature difference between the upper surface and lower surface of the sample.

Thermogravimetric (TG) experiments were conducted on a TA instruments (model Q50, TA Instruments, Newcastle, DE, USA) with a nitrogen flow rate of 40 mL/min. Samples (~2 mg) were heated in Al_2_O_3_ pans from 40 to 700 °C at a heating rate of 20 °C/min. The onset decomposition temperature, *T*_5%_, at which 5 wt% of the original weight was lost, and *T*_max_, at which the products possessed the maximum weight loss rate, were recorded together with the residue weight.

The morphologies of the HFFR-TC-PA6 composites were obtained from fracture surfaces from the impact test by using a scanning electron microscope (SEM, HITACHI S-550, Tokyo, Japan) operated at a 10 kV accelerating voltage. Samples were sputter-coated with gold before test.

Differential scanning calorimeter (DSC) analysis was carried out on DSC-60A, Shimadzu instruments. The specimen was heated from 40 to 260 °C at a heating rate of 10 °C/min, held for 3 min, cooled to 40 °C at a rate of 10 °C/min, and then finally reheated to 260 °C at a heating rate of 10 °C/min. The degrees of crystallinity of PA6 in the resulting composites were calculated as follows:(2)$$X_{c}=\frac{\Delta H_{m}}{p_{i}\cdot \Delta H_{m}^{0}}$$
where Δ*H_m_* is the melting enthalpy, obtained from melting peak of second heating curve of PA6, *p_i_* is weight percent of PA6, and $\Delta H_{m}^{0}$ is 100% crystalline enthalpy of PA6, which is 230 J/g [20].

## 3. Results

### 3.1. Effect of TC Fillers on the Mechanical Properties and Thermal Conductivity of HFFR-TC-PA6 Composites

The mechanical properties of HFFR-TC-PA6 with different AlN amounts are summarized in Table 2. These are averaged values along with standard deviations of five samples. As shown in Table 2, the tensile strength, flexural strength, and notched Izod impact strength of HFFR-TC-PA6 gradually decreased as the amount of AlN increased. The tensile strength, flexural strength, and notched Izod impact strength for the S0 composite were 61.16 MPa, 98.50 MPa and 8.99 KJ/m^2^, respectively, which decreased down to 37.51 MPa, 63.84 MPa and 6.11 KJ/m^2^, respectively, when 50 wt% AlN-550RFS was incorporated, thus presenting reductions of 38.67%, 35.19%, and 32.04%, respectively. Such reductions in mechanical properties could be primarily attributed to the poor compatibility between AlN particles and PA6, resulting in an uneven dispersion or agglomeration of particles in the polymer matrix and weak microstructures, which is beneficial for stress concentration, crack generation and propagation. The flexural modulus, however, gradually increased with increasing AIN amount, which was 2840 MPa for the S0 composite and presented up to a 58.91% enhancement when the 50 wt% AlN-550RFS was incorporated. Such an increase in flexural modulus may have been due to the rigidity of fillers, which improve the overall flexural resistance of composites. A similar trend was also observed in the HFFR-TC-PA6 composites filled with AlN-300SFS.

It is well known that the performance of material largely depends on the microstructures. The pristine AlN-550RFS and AlN-300SFS, and the fracture surfaces of the HFFR-TC-PA6 composites were analyzed using SEM, as shown in Figure 1. As seen in Figure 1a,b, the raw AlN-550RFS has a relatively regular spherical morphology, while the raw AlN-300SFS presented irregular shapes. In Figure 1c, the S0 composite without AlN had a rocky pattern fracture surface, thus indicating a fast fracture mechanism and a brittle fracture. Figure 1d,e shows the micrograph of the 30% AlN-550RFS and 30% AlN-300SFS composites, respectively. It can be observed that AlN particles had a weak filler-matrix interface, and voids left by the separation of the filler particles from the matrix resin could be observed. Figure 1f,g presents the micrograph of 50% AlN-550RFS and 50% AlN-300SFS, respectively, which shows that a large number of AlN particles fell off from the PA6 matrix during the impact test, and the gap between the filler and the PA6 matrix resin was obvious, thus indicating a poor compatibility. Furthermore, it can also be seen from Table 2 that with the same filler amount, the HFFR-TC-PA6 composite with AlN-550RFS had better mechanical properties than the HFFR-TC-PA6 composite with AlN-300SFS. The reason for this may be that the AlN-300SFS, which has a smaller particle size, tends to more easily aggregate, which results in the increase of defects in the composites and further leads to a decrease of mechanical properties [38].

A comparative analysis of the data in Table 2 shows that the thermal conductivity was a function of the weight fraction of AlN in the HFFR-PA6 composites. The thermal conductivity increased along with the increase of the AlN-550RFS amount. The thermal conductivity of 30% AlN-550RFS was 0.468 W/mK, similar to that (~0.5 W/mK) reported in the literature for the PA6/AlN composite with 30 wt% AlN [16], and this reveals the reliability and repeatability of the experiments. When the AlN-550RFS content was 50 wt%, the thermal conductivity of the HFFR-TC-PA6 composite increased to 0.739 W/mK, which was 238% higher than that of the S0 composite. It can be seen from Figure 1d,e that isolated particles were dispersed in the PA6 matrix when 30 wt% AlN-550RFS was incorporated, while overlapped particles were observed for the HFFR-TC-PA6 composite with 50 wt% AlN-550RFS, which formed connected paths for thermal transfer, resulting in a higher thermal conductivity. Compared to AlN-300SF, AlN-S550RFS presented a lower thermal conductivity due to a smaller contact area between the spherical AlN, which had more difficulty forming a conductive path than other forms of AlN [39].

The effects of two different types of BN fillers (BN-NW04 and BN-SW08) on the mechanical properties and thermal conductivity of the HFFR-TC-PA6 composite were also evaluated, as summarized in Table 3. The tensile and flexural strengths of HFFR-TC-PA6 composites filled with BN-NW04 (particle size was 0.5 µm) slightly decreased with the increase of the BN-NW04 amount from 30 to 50 wt%. On the contrary, the tensile strengths of the HFFR-TC-PA6 composites filled with BN-SW08 (particle size was 6 µm) were equivalent for both 30 and 50 wt% BN-SW08 with the uncertainty of error, while the composite with 50 wt% BN-SW08 presented a 10.53% higher flexural strength. The HFFR-TC-PA6 composite filled with 30 wt% BN-NW04 showed significantly higher tensile and flexural strengths as well as thermal conductivity than the HFFR-TC-PA6 composite filled with 30 wt% BN-SW08. This may have been induced by much smaller particle sizes, which increased the interfacial contacting area between the matrix and fillers, thus reducing the stress concentration or improving thermal transfer efficiency [40]. However, 50 wt% of BN particles with different particle sizes did not present obvious differences in tensile strength and thermal conductivity, which may be attributed to over-concentrated BN loadings that diminished the influence of particle size. The morphology of two types of BN and fracture surfaces of the HFFR-TC-PA6 composites are presented in Figure 2. As shown in Figure 2a,d, both BN particles had similar circular flake-like shapes but different sizes, i.e., an average particle size of 6 µm for BN-SW08 and 0.5 µm for BN-NW04. The SEM morphologies of composites illustrate that all the BN particles were well dispersed without obvious aggregation observed.

### 3.2. Effect of TC Fillers on Flame Retardancy of HFFR-TC-PA6 Composites

To evaluate the effects of different TC fillers on the flame retardancy for HFFR-TC-PA6, LOI and vertical burning (UL-94) tests were conducted, and the data are summarized in Table 4. The LOI values increased from 24.5% to 30% when AlN-550RFS was incorporated from 0% to 40 wt% in the HFFR-TC-PA6 composites. Figure 3 illustrates the digital photos of the burning residue after the LOI test. For the S0 composite, the burning area of the sample was much larger than composites with AlN-550RFS which form obvious smaller and compacter char after the LOI test. The vertical burning test (UL-94) measures the flammability and flame spread of plastic materials exposed to a small flame [41]. During the test, a V-0 rating can be achieved when the materials extinguished in less than 10 s after both the first and second flame implementations, which requires the flame retardant to work in an instant period [19,42]. In the UL-94 test, the S0 composite without the TC fillers passed the V-2 rating, composites containing 10–20 wt% AlN-550RFS passed either the V-0 rating for 3.2 mm-thick specimens or the V-2 rating for 1.6 mm-thick specimens. When the AlN-550RFS content increased to 30 wt% or higher, the V-0 rating was also obtained for 1.6 mm specimens. The burning residues of the samples after the UL-94 tests are shown in Figure 4.No obvious char layer could be observed for the S0 composite due to the dripping mechanism, which was similar for the 1.6 mm thick specimens of composites with 10~20 wt% AlN-550RFS [38]. As shown in Figure 4d,e, composites with 30, 40 and 50 wt% AlN-550RFS presented obvious car layers that were also slowly burnt during the test. After combining the results of the LOI and UL-94 tests, it can be seen that the addition of AlN-550RFS is beneficial for the improvement the flame retardancy of the HFFR-TC-PA6 composites. Figure 5 shows the char residue morphology of composites after the UL-94 test. As seen in Figure 5a, there was almost no char layer for the S0 composite, and a large area of the bare polymer matrix could be seen. On the contrary, it can be observed that the surface morphology of the 30% AlN-550RFS and 50% AlN-550RFS composites presented uniform and compact char layers. This suggests that the incorporation of AlN-550RFS could deposited on the combustion interface, which could effectively prevents heat and flammable gas transfer and consequently enhance the flame retardancy of the HFFR-TC-PA6 composites.

The LOI and vertical burning (UL-94) test results for the composites filled with AlN-300SFS, BN-SW08 and BN-NW04 are also summarized in Table 4. For all these composites, the LOI values increased with the increase of filler amount from 30 to 50 wt%. Compared with composites with AlN, the LOI values for composites with BN seemed to be more influenced by BN amount, which as they presented significant enhancement when BN increased from 30 to 50 wt%. All the composites filled with BN passed the V0 rating. Similar to AlN-550RFS, all the composites filled with AlN-300SFS, BN-SW08 and BN-NW04 presented obvious char layers, as shown in Figure 6.

### 3.3. Thermal Analysis

In order to understand the effects of TC fillers on the thermal stability and decomposition behaviors of the HFFR-TC-PA6 composites, thermogravimetric analyses (TGA) were carried out. The mass loss curves of the AlPi, TC fillers and HFFR-TC-PA6 composites in nitrogen atmosphere are presented in Figure 7, Figure 8 and Figure 9. The relevant thermal decomposition data are summarized in Table 5. As shown in Figure 7a, all four TC fillers remained unchanged in the tested temperature range, indicating a high thermal stability [43]. It can be seen from Table 5 that the initial decomposition temperature (T_5%_) of neat AlPi was 394 °C. According to previous reports, the initial decomposition temperature (T_5%_) of PA6 is about 400 °C [42]. With the incorporation of AlPi into the PA6 matrix, the initial decomposition temperature of PA6/AlPi (S0 composite) decreased to 377 °C. This phenomenon can be explained by the fact that AlPi catalyzed the degradation of PA6 and reduced the thermal stability of the S0 composite [44]. With the addition of AlN-550RFS, as shown in Figure 8 and Table 5, the initial decomposition temperature of the composites significantly increased. This could be attributed to the high thermal stability of AlN-550RFS. As presented in Figure 8b, the incorporation of AlN-550RFS remarkably decreased the maximal mass loss rate, and the rate of mass loss rate was further decreased with the increase of AlN-550RFS. As shown in Figure 9, the initial decomposition temperature of the composite with 30 wt% BN-SW08 reduced from the 377 °C of the S0 composite to 352 °C, indicating that the incorporation of BN-SW08 into FRPA6 stimulated the decomposition of FRPA6 at an early stage. However, composites with either AlN-300SFS or BN-NW04 had similar thermal behaviors to composites with AlN-550RFS. Additionally, all TC fillers rarely changed the peak temperatures of composites—around 450 °C.

### 3.4. Flammability Behavior

A cone calorimeter based on the oxygen consumption principle is a useful instrument to evaluate the combustion performance of polymeric materials. In order to clearly understand the effect of the TC filler on the flame retardancy of HFFR-TC-PA6, cone calorimetric analyses were performed. The data are summarized in Table 6, including time-to-ignition (TTI), peak heat release rate (PHRR), time of peak heat release rate (t_PHRR_), total heat release (THR), total smoke rate (TSR), and mean effective heat combustion (MEHC), along with deduced quantities of the maximum value of the average rate of heat emission (MARHE). Figure 10 illustrates the heat release rate (HRR) and total heat release (THR) curves of all the samples.

The TTI is used to determine the influences of a filler or flame retardant on ignitability, which can be measured from the onset on an HRR curve [19]. As listed in Table 6, the presence of AlN-550RFS increased the TTI, while AlN-550RFS mainly enhanced the complex viscosity of composites and effectively retarded the heat diffusion into the PA6 matrix and the release of combustion gas [45]. Such phenomenon has also been observed in previous work [46,47].

HRR measures the heat release per unit surface area of a burning specimen, which is considered to have a significant influence on fire hazard [48]. On the other hand, PHRR is regarded as an important parameter for the assessment of a real fire hazard [49]. As shown in Table 6, the PHRR value for the S0 composite was 797 kW/m^2^, which decreased down to 520 (34.8%) and 451 kW/m^2^ (43.4%) for 30% AlN-550RFS and 50% AlN-550RFS composites, respectively. The HRR curves of the S0 and HFFR-TC-PA6 composites with varied AlN-550RFS amounts are presented in Figure 10a. It can be seen that both the S0 and 10% AlN-550RFS composites presented higher heat release rates and shorter overall combustion times, indicating a fierce burning, while higher AlN-550RFS amounts presented a lower heat release rate and longer overall combustion times. To evaluate the fire hazard of the samples, MAHRE was introduced. The MAHRE values for the HFFR-TC-PA6 composites decreased with increasing AlN-550RFS amounts, suggesting an enhanced fire safety. Furthermore, the dependence of the THR curves on the AlN-550RFS amount was not obvious, as shown in Figure 10b.

The MEHC value reveals the burning rate of volatile gases in gaseous phase flame during combustion, where a lower MEHC value indicates the exerting flame retardant effect in the gaseous phase [10]. As shown in Table 6, the MEHC values of the HFFR-TC-PA6 composites were similar to that of the S0 composite, indicating that the gaseous phase combustion was basically not affected by AlN-550RFS.

Moreover, the effect of the TC filler types on the cone calorimetric performance of the HFFR-TC-PA6 composites was also investigated. As shown in Table 7, the TTI of 30% AlN-550RFS and 30% AlN-300RFS were similar but obviously lower than those of 30% BN-SW08 and 30% BN-NW04. This may have been due to the higher complex viscosity of the BN-filled HFFR-TC-PA6 composites. Figure 11a reflects the HRR curves of the S0 and HFFR-TC-PA6 composites that carried TC fillers. As it can be seen that all the TC fillers significantly reduced the PHRR values, with the lowest value, 309 kW/m^2^, observed for the 30% BN-SW08 composite.

### 3.5. DSC Analyses of the Composites

The crystallization behavior of the S0 and HFFR-TC-PA6 composites were investigated using DSC, as presented in Figure 12 and summarized in Table 8. The crystallization temperature (T_c_) of S0 was 185.5 °C, which increased by 4.2 to 7.1 °C with the incorporation of TC fillers, and the increasement was influenced by filler type and filler amount. However, the crystallization (X_c_) and melting temperature of S0 did not appear to be influence by filler type and filler amount, which were around 20% and 220 °C, respectively.

## 4. Conclusions

Four types of TC fillers have been reported that have an obvious flame-retardant efficiency for the PA6/AlPi composites. The specimen with a thickness of 1.6 mm can pass the UL-94 V-0 flammability rating, and its LOI values exceeded 29% when the loading of TC fillers was more than 30 wt%. The incorporation of TC fillers promoted the formation of an integrated char layer. The thermal conductivity of the HFFR-TC-PA6 composite with 50 wt% BN-SW08 was 0.93 W/mK, which was 200% more than that of the composite without TC fillers. The thermal stability of the composites was improved with the addition of thermal conductive fillers. Finally, these thermal conductive fillers were found to play an important role in heterogeneous nucleation, resulting higher crystallization temperatures.

## Figures and Tables

**Figure 1 materials-12-04114-f001:**
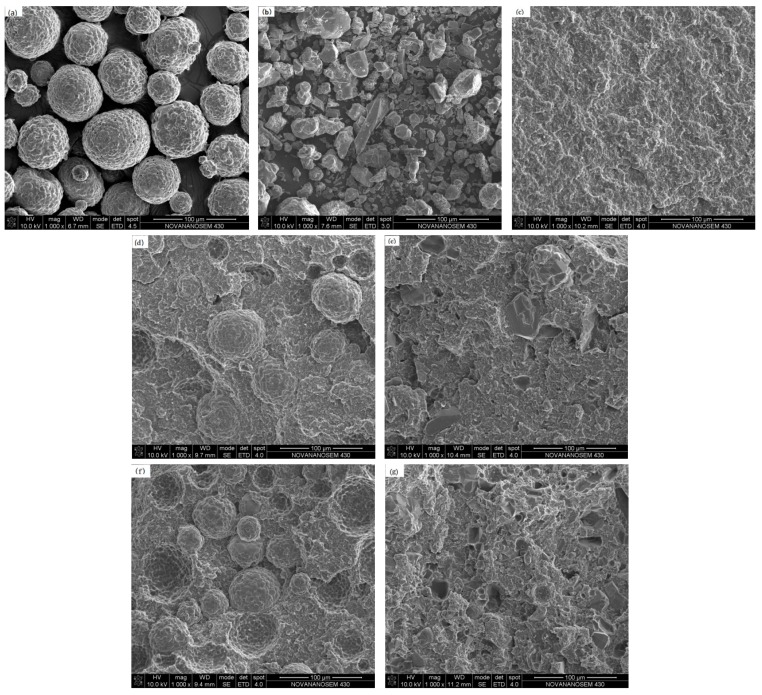
SEM micrographs of (**a**) AlN-550RFS, (**b**) AlN-300SFS, (**c**) S0, (**d**) 30% AlN-550RFS composites, (**e**) 30% AlN-300SFS, (**f**) 50% AlN-550RFS composites, and (**g**) 50% AlN-300SFS composites.

**Figure 2 materials-12-04114-f002:**
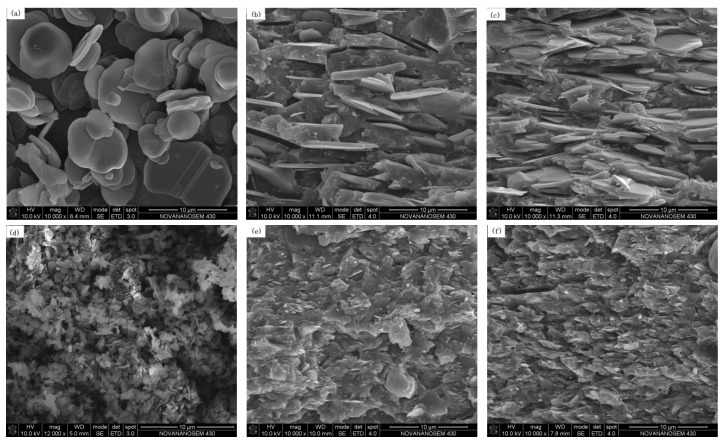
SEM images of (**a**) raw BN-SW08, (**b**) 30% BN-SW08, (**c**) 50% BN-SW08, (**d**) raw BN-NW04, (**e**) 30% BN-NW04, and (**f**) 50% BN-NW04.

**Figure 3 materials-12-04114-f003:**
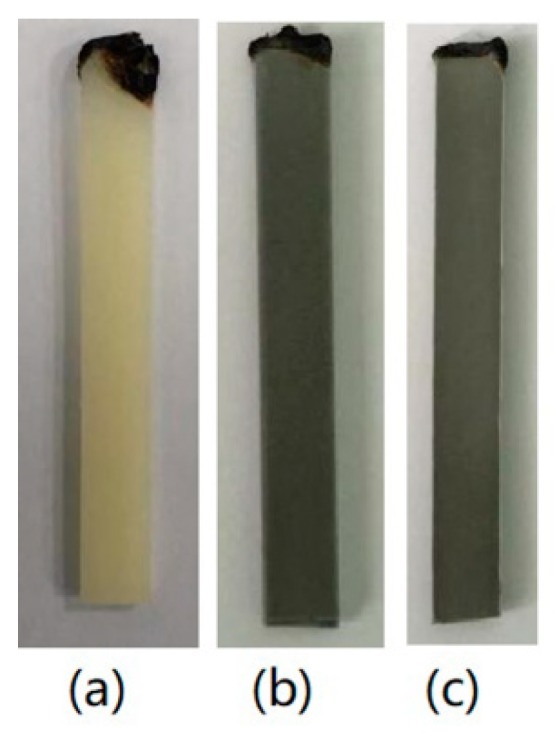
Sample morphology photograph after limiting oxygen index (LOI) tests: (**a**) S0; (**b**) 30% AlN-550RFS; and (**c**) 50% AlN-550RFS.

**Figure 4 materials-12-04114-f004:**
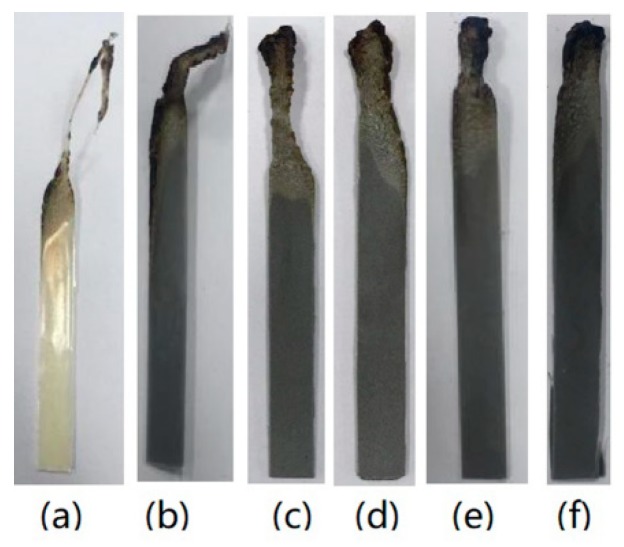
Char layer of the samples with a thickness of 1.6 mm after the UL-94 (vertical burning test) tests: (**a**) S0, (b) 10% AlN-550RFS, (**c**) 20% AlN-550RFS, (**d**) 30% AlN-550RFS, (**e**) 40% AlN-550RFS, and (**f**) 50% AlN-550RFS.

**Figure 5 materials-12-04114-f005:**
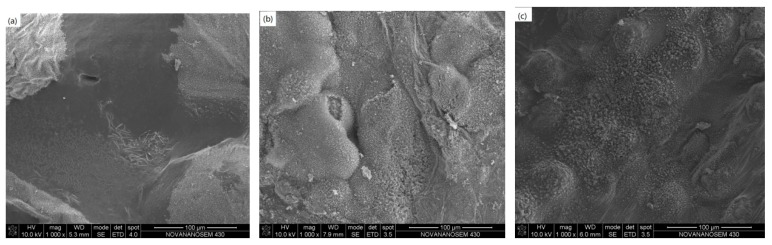
SEM photographs of char layer after the UL-94 test of (**a**) S0, (**b**) 30% AlN-550RFS, and (**c**) 50% AlN-550RFS.

**Figure 6 materials-12-04114-f006:**
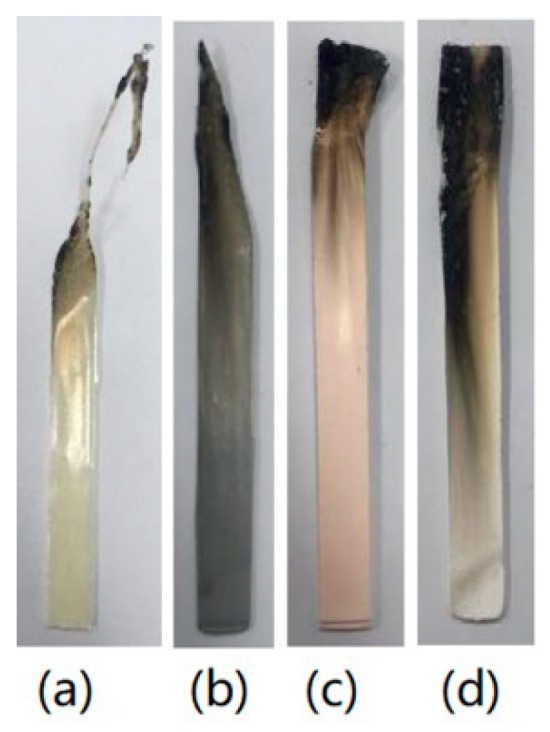
Char layer of the samples with thicknesses of 1.6 mm after the UL-94 tests: (**a**) S0, (**b**) 30% AlN-300SFS, (**c**) 30% BN-SW08, and (**d**) 30% BN-NW04.

**Figure 7 materials-12-04114-f007:**
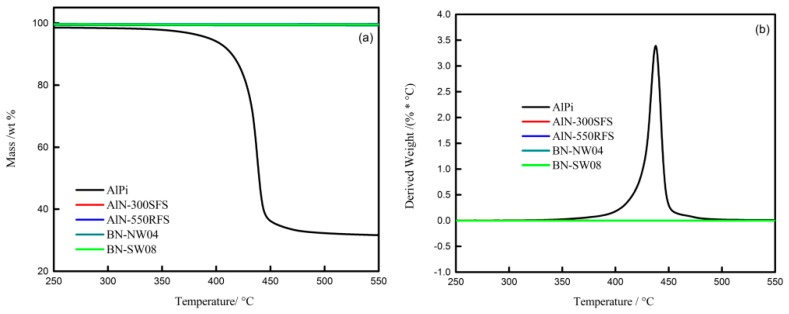
Thermogravimetric (TG) (**a**) and Derivative Thermogravimetry (DTG) (**b**) curves of AlPi, AlN-300SFS, AlN-550RFS, BN-SW08 and BN-NW04 under N_2_.

**Figure 8 materials-12-04114-f008:**
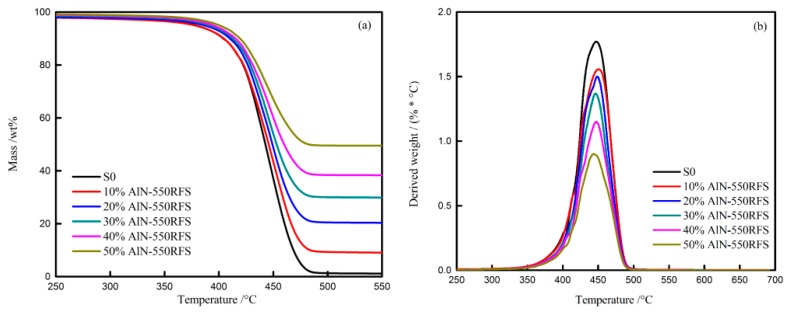
TG (**a**) and DTG (**b**) curves of the HFFR-TC-PA6 composites with different AlN-550RFS ratios under N_2_.

**Figure 9 materials-12-04114-f009:**
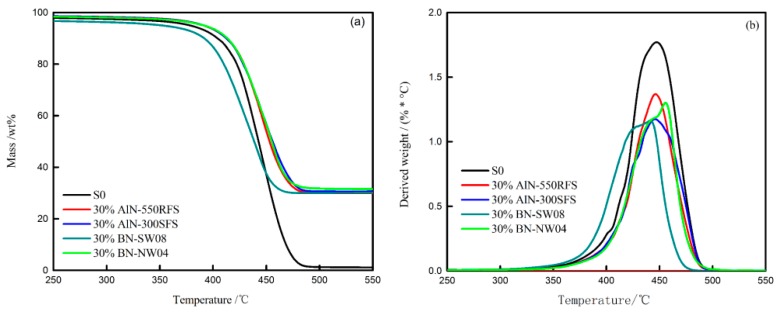
TG (**a**) and DTG (**b**) curves of the HFFR-TC-PA6 composites with different TC fillers under N_2_.

**Figure 10 materials-12-04114-f010:**
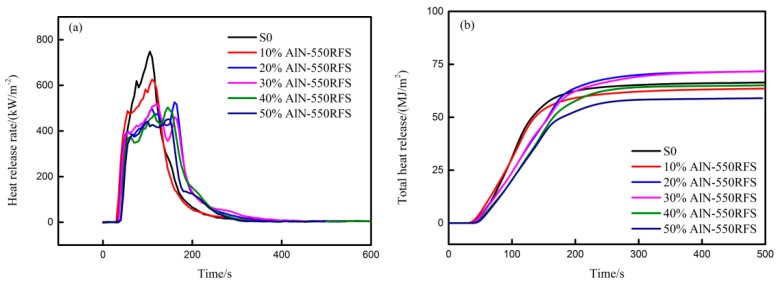
Cone calorimetric results of the experimental samples at an external heat flux of 50 kW/m^2^ as a function of burning duration: (**a**) heat release rate (HRR) curves and (**b**) total heat release (THR) curves.

**Figure 11 materials-12-04114-f011:**
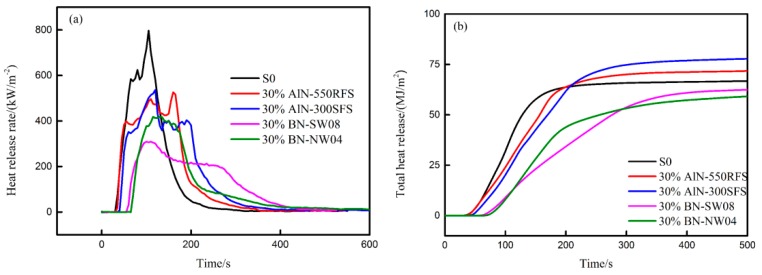
Cone calorimetric results of the experimental samples at an external heat flux of 50 kW/m^2^ as a function of burning duration: (**a**) HRR curves and (**b**) THR curves.

**Figure 12 materials-12-04114-f012:**
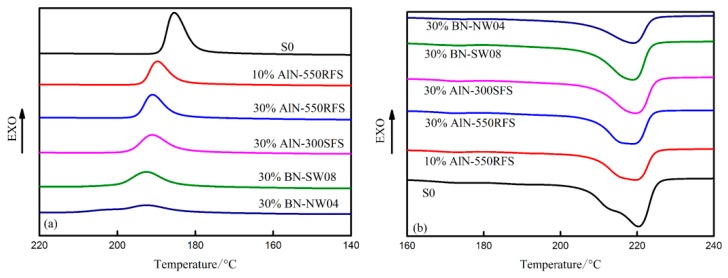
(**a**) Differential scanning calorimeter (DSC) cooling curves of the HFFR-TC-PA6 composites and (**b**) the second heating curves of the HFFR-TC-PA6 composites.

**Table 1 materials-12-04114-t001:** Mixing ratios of different materials in composites.

Name of Composite	PA6/AlPi/TC Filler	TC Filler Content (wt%)
S0	90:10:0	0%
10% AlN-550RFS	90:10:11.11	10%
20% AlN-550RFS	90:10:25	20%
30% AlN-550RFS	90:10:42.86	30%
40% AlN-550RFS	90:10:66.67	40%
50% AlN-550RFS	90:10:100	50%
30% AlN-300SFS	90:10:42.86	30%
50% AlN-300SFS	90:10:100	50%
30% BN-SW08	90:10:42.86	30%
50% BN-SW08	90:10:100	50%
30% BN-NW04	90:10:42.86	30%
50% BN-NW04	90:10:100	50%

**Table 2 materials-12-04114-t002:** Mechanical properties and thermal conductivity of the halogen-free flame retardant (HFFR) and thermal conductive (TC) polyamide 6 (PA6) (HFFR-TC-PA6) composite with different amounts of spherical aluminum nitride (AlN)-550RFS and non-spherical AlN-300SFS.

Name of Composite	Tensile Strength (MPa)	Flexural Strength (MPa)	Flexural Modulus (MPa)	Notched Izod Impact (KJ/m^2^)	Thermal Conductivity (W/mK)
S0	61.16 ± 1.94	98.50 ± 4.67	2840 ± 73	8.99 ± 0.15	0.310
10% AlN-550RFS	55.61 ± 0.60	94.24 ± 3.07	3151 ± 68	7.42 ± 0.4	0.365
20% AlN-550RFS	49.85 ± 1.47	90.26 ± 0.47	3282 ± 77	7.03 ± 0.16	0.404
30% AlN-550RFS	46.04 ± 0.64	88.99 ± 2.90	3911 ± 151	6.77 ± 0.13	0.468
40% AlN-550RFS	41.78 ± 1.38	85.93 ± 1.15	4438 ± 852	6.44 ± 0.3	0.572
50% AlN-550RFS	37.51 ± 0.89	63.84 ± 0.17	4513 ± 188	6.11 ± 0.09	0.739
30% AlN-300SFS	40.54 ± 0.11	73.99 ± 1.98	3207 ± 135	7.01 ± 0.41	0.505
50% AlN-300SFS	33.50 ± 0.81	64.33 ± 1.03	4457 ± 279	5.31 ± 0.13	0.799

**Table 3 materials-12-04114-t003:** Mechanical properties and thermal conductivity of HFFR-TC-PA6 with different boron nitride (BN) amounts.

Name of Composite	Tensile Strength (MPa)	Flexural Strength (MPa)	Flexural Modulus (MPa)	Notched Izod Impact (KJ/m^2^)	Thermal Conductivity (W/mK)
30% BN-SW08	48.62 ± 0.97	92.17 ± 0.33	6065 ± 360	7.60 ± 0.34	0.567
50% BN-SW08	49.78 ± 1.08	101.88 ± 3.35	6567 ± 402	No data*	0.930
30% BN-NW04	58.65 ± 0.69	96.67 ± 1.24	5469 ± 137	4.79 ± 0.22	0.690
50% BN-NW04	51.14 ± 0.95	93.65 ± 0.54	8651 ± 570	No data*	0.910

No data*: The samples are too brittle to get notched Izod impact data.

**Table 4 materials-12-04114-t004:** Flame retardancy of HFFR-TC-PA6 with different TC fillers.

Name of Composite	Flame Retardancy				
	LOI, %	UL94 (3.2 mm)	Dripping (3.2 mm)	UL94 (1.6 mm)	Dripping (1.6 mm)
S0	24.5	V2	No	V2	Yes
10% AlN-550RFS	26	V0	No	V2	Yes
20% AlN-550RFS	26.5	V0	No	V2	Yes
30% AlN-550RFS	29	V0	No	V0	No
40% AlN-550RFS	30	V0	No	V0	No
50% AlN-550RFS	30	V0	No	V0	No
30% AlN-300SFS	28	V0	No	V2	Yes
50% AlN-300SFS	29.5	V0	No	V0	No
30% BN-SW08	31	V0	No	V0	No
50% BN-SW08	40	V0	No	V0	No
30% BN-NW04	30	V0	No	V0	No
50% BN-NW04	37	V0	No	V0	No

**Table 5 materials-12-04114-t005:** TG data of the HFFR-TC-PA6 composites’ TC fillers ratios.

Name of Composite	T_5%_ (°C)	T_max_ (°C)	The Char Residues at 700 °C (%)
AlPi	394	434	31.7
S0	377	447	1.2
10% AlN-550RFS	377	450	8.8
20% AlN-550RFS	386	448	20.3
30% AlN-550RFS	391	447	29.9
40% AlN-550RFS	395	447	38.6
50% AlN-550RFS	400	445	49.5
30% AlN-300SFS	391	446	30.7
30% BN-SW08	352	441	30.8
30% BN-NW04	391	455	31.5

**Table 6 materials-12-04114-t006:** Cone calorimetric data of the testing samples with a heat flux of 50 KW/m^2^.

Name of Composite	TTI(s)	PHRR (KW/m^2^)	t_PHRR_(s)	THR (MJ/m^2^)	TSR (m^2^/m^2^)	MEHC (MJ/kg)	MAHRE ^a^ (KW/m^2^)
S0	34	797	105	67	945	26.2	393
10% AlN-550RFS	36	625	110	64	876	25.8	368
20% AlN-550RFS	39	535	120	77	1055	25.1	339
30% AlN-550RFS	45	520	120	71	938	25.6	327
40% AlN-550RFS	43	503	145	66	790	26.1	304
50% AlN-550RFS	43	451	150	60	732	25.8	287

^a^ AHRE is defined as average rate of heat emission which is calculated from the division of cumulative heat emission and time; MAHRE denotes the maximum value of AHRE.

**Table 7 materials-12-04114-t007:** Cone calorimetric data of the testing samples with a heat flux of 50 KW/m^2^.

Name of Composite	TTI(s)	PHRR (KW/m^2^)	t_PHRR_(s)	THR (MJ/m^2^)	TSR (m^2^/m^2^)	MEHC (MJ/kg)	MAHRE (KW/m^2^)
S0	34	797	105	67	945	26.2	393
30%AlN-550RFS	45	520	120	71	938	25.6	327
30%AlN-300RFS	46	536	120	79	1145	26.0	310
30%BN-SW08	62	309	100	63	896	25.6	181
30%BN-NW04	70	418	115	61	738	26.2	221

**Table 8 materials-12-04114-t008:** DSC results of the HFFR-TC-PA6 composites.

Name of Composite	T_C_ (°C)	T_m_ (°C)	ΔH_m_ (J/g)	X_c_ (%)
S0	185.5	220.3	41.9	20.2
10% AlN-550RFS	189.7	219.6	36.1	19.4
30% AlN-550RFS	191.1	218.8	31.7	21.9
30% AlN-300RFS	191.0	219.6	31.7	21.9
30% BN-SW08	192.6	218.9	30.8	21.2
30% BN-NW04	192.6	219.0	30.1	20.6

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
