# Peer review of "Effect of Thermal Conductive Fillers on the Flame Retardancy, Thermal Conductivity, and Thermal Behavior of Flame-Retardant and Thermal Conductive Polyamide 6"

_materials, 2019, doi:10.3390/ma12244114_

Round 1
Reviewer 1 Report
This paper deals with the interesting topic of using nylon for preparation of flame-retardant material and exploring its thermal and mechanical behavior due to nylon’s diverse range of applications. The submitted work can be reconsidered for publication at Materials if the following points are fully addressed in a revised version of the manuscript.
Introduction of the manuscript need some more data from latest reviews to clearly make a background for users to understand the work, with special focus on articles dealing with the use of nylon in FDM processes. What is the extent of the dispersion of fillers in matrix material? How the greater level of the dispersion can be ensured with the particular technique? Please outline some of the possible application for the material under consideration. Briefly explain the correlation between the decomposition and flame retardant property of the material. In section 2.1, it has been claimed that PA6 was dried at 100℃ for 24 hours but in section 2.2 the temperature is claimed to be 80℃. Please correct this with accurate data. In manuscripts rather than using “thermal conductive filler” authors are suggested to use nominations for fillers, such as composition. ISO 527: 2012 has been withdrawn, please replace with latest version of the standard i.e., ISO 527-1:2019. Additionally, authors are suggested to check all other standards used in this research work. In section 3.4, please check 2nd sentence for grammatical mistake. Section 3.4 and 3.5, is filled with the wrong cross reference for tables and figures. For example, Data obtained from cone calorimeter are summarized in Table 6, but authors claim that data to be in table 8, which is not correct. Please check for these discrepancies. Finally, the English should be improved as well; I would recommend double checking the text before re-submitting, possibly with the help of a mother-tongue reader
Author Response
The following content is the same as the attachment.
Response to Reviewer1 comments
Dear Reviewer:
Thank you very much for your efficient work and the comments about our manuscript entitled “Effect of Thermal Conductive Fillers on the Flame retardancy, Thermal Conductivity, and Thermal Behavior of Flame-Retardant and Thermal Conductive Polyamide 6” (No.: materials-628424).
We have checked the manuscript carefully and revised it according to the your comments and suggestions. Here we submit a new version of our manuscript. The following is a point-to-point response to the your comments.
Reviewer #1:
Comments to the Author
This paper deals with the interesting topic of using nylon for preparation of flame-retardant material and exploring its thermal and mechanical behavior due to nylon’s diverse range of applications. The submitted work can be reconsidered for publication at Materials if the following points are fully addressed in a revised version of the manuscript.
Comments on the manuscript:
Comment 1:
Introduction of the manuscript need some more data from latest reviews to clearly make a background for users to understand the work, with special focus on articles dealing with the use of nylon in FDM processes.
Response: Thank you for the comment. Agreed. Now we had checked the introduction of the manuscript carefully and rewritten the introduction part.
Comment 2:
What is the extent of the dispersion of fillers in matrix material?
Response: Thank you for the comment. About the extent of dispersion of fillers in matrix material, in addition to visual analysis using scanning or transmission electron microscopy, some researcher used the relationship between normalized storage modulus versus angular frequency as a means of characterization, the steeper the slope, the higher the dispersion degree. As described in reference [34].
Nikoo, G.; Armin, S. E. S.; Milad, M.; Hossein, N. The effect of filler localization on morphology and thermal conductivity of the polyamide/cyclic olefin copolymer blends filled with boron nitride. Journal Article.
Comment 3:
How the greater level of the dispersion can be ensured with the particular technique?
Response: Thank you for the comment. In order to obtain greater level of the dispersion, there were some methods had been researched. For example, improving the compatibility between the filler and the matrix resin with additives such as coupling agents[35], using the pre-dispersion technology to improve the dispersion of fillers[34], and regulating the dispersion state of the fillers using extrusion stretching, injection molding methods or other processing methods, and so on [36].
Comment 4:
Please outline some of the possible application for the material under consideration.
Response: Thank you for the suggestion. The HFFR-TC-PA6 can be used in electrical connectors, switch components, lamp holder for light emitting diodes (LEDs). Please see Line 46~47, 100~101.
Comment 5:
Briefly explain the correlation between the decomposition and flame retardant property of the material.
Response: Thank you for the commen. The heat resistance and flame retardancy of the polymer material are not equivalent. Some materials have good heat resistance and excellent flame retardancy, such as polyphenylene ether (PPO), polyimide(PI), and the like. However, some polymers have poor heat resistance but good flame retardancy, such as polyvinyl chloride(PVC). For nanocomposites, having a high residual carbon content and a low maximum mass loss temperature is an important mention for better flame retardant performance, it can be refer to the following paper:
Zhu J , Start P , Mauritz K A , et al. Thermal stability and flame retardancy of poly(methyl methacrylate)-clay nanocomposites[J]. Polymer Degradation & Stability, 2002, 77(2):253-258.
Gu A , Liang G . Thermal degradation behaviour and kinetic analysis of epoxy/montmorillonite nanocomposites[J]. Polymer Degradation & Stability, 2003, 80(2):383-391.
However, for polymer/flame retardant composites, the relationship between thermal stability and flame retardancy is related to the reactivity between the flame retardant and the polymer matrix under thermal conditions. Please see page 10, line 313~315.
Comment 6:
In section 2.1, it has been claimed that PA6 was dried at 100℃ for 24 hours but in section 2.2 the temperature is claimed to be 80℃. Please correct this with accurate data.
Response: Sorry for our incorrect description. We have correct and re-edited sections 2.1 and 2.2. Please see Page 3, section 2.1 and section 2,2.
Comment 7:
In manuscripts rather than using “thermal conductive filler” authors are suggested to use nominations for fillers, such as composition.
Response: Thank you for the suggestion Now the related description has been modified in several places in the article, such as the section 3.1.
Comment 8:
ISO 527: 2012 has been withdrawn, please replace with latest version of the standard i.e., ISO 527-1:2019. Additionally, authors are suggested to check all other standards used in this research work.
Response: Sorry for our incorrect description. We have corrected the error, and the latest version of the standard has been referenced in the text. Please see Page 3, section 2.3.
Comment 9:
In section 3.4, please check 2nd sentence for grammatical mistake. Section 3.4 and 3.5, is filled with the wrong cross reference for tables and figures. For example, Data obtained from cone calorimeter are summarized in Table 6, but authors claim that data to be in table 8, which is not correct. Please check for these discrepancies.
Response: Sorry for our incorrect description. The grammatical mistake of the 2nd sentence has been corrected. The wrong cross reference for tables and figures have been corrected too.
Comment 10:
Finally, the English should be improved as well.
Response: The English expression of this paper has been retouched by a Assistant Research fellow who has worked in the United States for more than 7 years.
All of the changes were marked red in the manuscript.
We appreciate for your warm work earnestly, and hope that the correction will meet with approval. Once again, thank you very much for your efficient work and your constructive comments.
Best wishes.
Sincerely yours,
Fang Wang, Wenbo Shi, Yuliang Mai, Wei Hu and Bing Liao

Reviewer 2 Report
The paper put forward by Fang Wang et.al entitled “Effect of Thermal Conductive Fillers on the Flame retardancy, Thermal Conductivity, and Thermal Behavior of Flame-Retardant and Thermal Conductive Polyamide 6” shows very important drawbacks preventing publication in its current form. The manuscript requires carefully, solid, reliable and in-depth improvement.
The biggest complaint that eliminates the possibility of publication is the lack of scientific novelty of the research topic. This is unacceptable. The authors should present a concise overview of the latest literature reports on their field of research and on this basis show the novelty of their results. Unfortunately, there is no proper state of the art and the novelty of the paper in certainly very low. Scientific novelty of the work have to be clarified and explained deeply in details.
Moreover, Materials and Methods selection is not properly justified. The description of the composites preparation processes is insufficient. In addition, there are many shortcomings regarding the research methods in the characterization section.
Furthermore, there are no characterization of both flame retardant and thermal conductive fillers, why? This would certainly raise the level of the article.
Only after a thorough change in the introduction and materials section and also the language level of the article, it is possible to continue the publication process of the article. I suggest review and resubmit.
Author Response
The following content is the same as the attachment.
Response to reviewer 2
Dear Reviewer:
Thank you very much for your efficient work and the comments about our manuscript entitled “Effect of Thermal Conductive Fillers on the Flame retardancy, Thermal Conductivity, and Thermal Behavior of Flame-Retardant and Thermal Conductive Polyamide 6” (No.: materials-628424).
We have checked the manuscript carefully and revised it according to the your comments and suggestions. Here we submit a new version of our manuscript. The following is a point-to-point response to the your comments.
Reviewer #2:
Comments to the Author
The paper put forward by Fang Wang et.al entitled “Effect of Thermal Conductive Fillers on the Flame retardancy, Thermal Conductivity, and Thermal Behavior of Flame-Retardant and Thermal Conductive Polyamide 6” shows very important drawbacks preventing publication in its current form. The manuscript requires carefully, solid, reliable and in-depth improvement.
Comments on the manuscript:
Comment 1:
The biggest complaint that eliminates the possibility of publication is the lack of scientific novelty of the research topic. This is unacceptable. The authors should present a concise overview of the latest literature reports on their field of research and on this basis show the novelty of their results. Unfortunately, there is no proper state of the art and the novelty of the paper in certainly very low. Scientific novelty of the work have to be clarified and explained deeply in details.
Response: Thank you for the comment very much. There have been a lot of research reports on halogen-free flame retardant PA6 (HFFR-PA6) [19, 25, 27~28......] or thermal conductive PA6 (TC) [20~21, 34......]. However, there are very few reports on the halogen free flame retardant and thermally conductive PA6 (HFFR-TC-PA6). But, in industrial applications, PA6 is often required to have both flame retardancy and thermal conductivity, such as LED bulb base, coil skeleton , connector housing, etc.. As a kind of filler, thermal conductive filler may affect the flame retardancy and mechanical properties of PA6 composites[37]. Therefore, it is of practical and academic significance to study the effect of thermally conductive fillers on the properties of HFFR-TC-PA6 composites. We have revisited the literature reports on the related fields and describe the meaning of our work in the section Introduction. Please see the rewritten Introduction section.
Comment 2:
Moreover, Materials and Methods selection is not properly justified. The description of the composites preparation processes is insufficient. In addition, there are many shortcomings regarding the research methods in the characterization section.
Response: Thank you for the comment. We have made modifications and additions to the Materials and Methods. A more specific description of the composites preparation processes has been made. The shortcomings in the research methods have been corrected.
Comment 3:
Furthermore, there are no characterization of both flame retardant and thermal conductive fillers, why? This would certainly raise the level of the article.
Response: Thank you for the comment. We have supplemented the relevant characterization of raw materials. Figure 1(a), Figure 1(b), Figure 2(a) and Figure 2(b) are the SEM micrographs for AlN-550RFS, AlN-300SFS, BN-SW08 and BN-NW04, respectively. Figure 7(a) and Figure 7(b) are the TG and DTG curves for the AlPi and four thermal conductive fillers.
All of the changes were marked red in the manuscript.
We appreciate for your warm work earnestly, and hope that the correction will meet with approval. Once again, thank you very much for your efficient work and your constructive comments.
Best wishes.
Sincerely yours,
Fang Wang, Wenbo Shi, Yuliang Mai, Wei Hu and Bing Liao

Reviewer 3 Report
There is no information as to whether the additives used were also dried. Two different drying temperature values are indicated (100oC and 80oC - lines 77 and 86. No specific nitrogen flow value during TGA measurement. How the thermal conductivity of the composites were determined - add information in the methodology The sample geometry used in mechanical tests was not given. No TGA results for used additives. The authors conducted TGA measurements in an inert gas N2 atmosphere (pyrolytic process), where other decomposition processes occur. In this case, it would be more appropriate to conduct TGA tests in an air atmosphere.
Author Response
The following content is the same as the attachment.
Response to reviewer 3
Dear Reviewer:
Thank you very much for your efficient work and the comments about our manuscript entitled “Effect of Thermal Conductive Fillers on the Flame retardancy, Thermal Conductivity, and Thermal Behavior of Flame-Retardant and Thermal Conductive Polyamide 6” (No.: materials-628424).
We have checked the manuscript carefully and revised it according to the your comments and suggestions. Here we submit a new version of our manuscript. The following is a point-to-point response to the your comments.
Reviewer #3:
Comments to the Author
There is no information as to whether the additives used were also dried. Two different drying temperature values are indicated (100oC and 80oC - lines 77 and 86. No specific nitrogen flow value during TGA measurement. How the thermal conductivity of the composites were determined - add information in the methodology The sample geometry used in mechanical tests was not given. No TGA results for used additives. The authors conducted TGA measurements in an inert gas N2 atmosphere (pyrolytic process), where other decomposition processes occur. In this case, it would be more appropriate to conduct TGA tests in an air atmosphere.
Comments on the manuscript:
Comment 1:
There is no information as to whether the additives used were also dried.
Response: Thank you for the comment. We have added the drying conditions for additives in Section 2.2.
Comment 2: Two different drying temperature values are indicated (100 °C and 80 °C - lines 77 and 86.
Response: Thank you for the comment. The error has been have corrected, Please see Section 2.1 and 2.2.
Comment 3:
No specific nitrogen flow value during TGA measurement.
Response: Thank you for pointing this fault. The flow value of the nitrogen is 40mL/min and this has been added in Page 4, line 157.
Comments 4:
How the thermal conductivity of the composites were determined - add information in the methodology
Response: Thank you for pointing this fault. The test method of the thermal conductivity has been added in Page 4, line 143~155.
Comments 5:
No TGA results for used additives.
Response: Thank you for pointing this fault. We have added the TGA results for used additives, Please see Figure 7(a), Figure 7(b) and Table 5.
Comments 6:
The authors conducted TGA measurements in an inert gas N2 atmosphere (pyrolytic process), where other decomposition processes occur. In this case, it would be more appropriate to conduct TGA tests in an air atmosphere.
Response: Thank you for the comment. We have referred to some literatures and found that in the study of the relationship between flame retardancy and thermal stability, many of the TG tests used a nitrogen atmosphere, so we also chose a nitrogen atmosphere. But according to your suggestion, we also arranged a TG-DSC test under the air atmosphere, the results are listed in the attachment file.
We found that the degradation tendency of the composites under air atmosphere is similar to that under nitrogen atmosphere, but the initial decomposition temperature(T5%) and residual carbon ratio are higher than those of under the nitrogen atmosphere, indicating that the solid phase flame retardant mechanism is strengthened under the condition of oxygen. Considering that the additives were also tested under a nitrogen atmosphere, so this part of the air test results are not included in the article.
All of the changes were marked red in the manuscript.
We appreciate for your warm work earnestly, and hope that the correction will meet with approval. Once again, thank you very much for your efficient work and your constructive comments.
Best wishes.
Sincerely yours,
Fang Wang, Wenbo Shi, Yuliang Mai, Wei Hu and Bing Liao

Reviewer 4 Report
- The results for 30%AlN-300SFS and 50%AlN-300SFS should be shift from Table 3 to Table 2
- Authors should extend description about mechanical properties. Higher mechanical strength of 30%BN-NW04 in comparison to 30%BN-NW08 can be also related to size of the fillers. It would be also interesting to consider the surface area of the fillers.
- line 203 – additional space
Author Response
The following content is the same as the attachment.
Response to reviewer 4
Dear Reviewer:
Thank you very much for your efficient work and the comments about our manuscript entitled “Effect of Thermal Conductive Fillers on the Flame retardancy, Thermal Conductivity, and Thermal Behavior of Flame-Retardant and Thermal Conductive Polyamide 6” (No.: materials-628424).
We have checked the manuscript carefully and revised it according to the your comments and suggestions. Here we submit a new version of our manuscript. The following is a point-to-point response to the your comments.
Reviewer #4:
Comments to the Author
- The results for 30%AlN-300SFS and 50%AlN-300SFS should be shift from Table 3 to Table 2.
- Authors should extend description about mechanical properties. Higher mechanical strength of 30%BN-NW04 in comparison to 30%BN-NW08 can be also related to size of the fillers. It would be also interesting to consider the surface area of the fillers.
- line 203 – additional space
Comments on the manuscript:
Comment 1:
The results for 30%AlN-300SFS and 50%AlN-300SFS should be shift from Table 3 to Table 2.
Response: Thank you for the suggestion. We have shifted the results for 30%AlN-300SFS and 50%AlN-300SFS from Table 3 to Table 2. Please see Table 2 in Page 6 and Table 3 in Page 7.
Comment 2: Authors should extend description about mechanical properties. Higher mechanical strength of 30%BN-NW04 in comparison to 30%BN-SW08 can be also related to size of the fillers. It would be also interesting to consider the surface area of the fillers.
Response: Thank you for the comment. We have reanalyzed and described the mechanical properties, Please see the section 3.1 in page 5~7. We also analyzed the reason why higher mechanical strength of 30%BN-NW04 in comparison to 30%BN-SW08,and we agree with your opinion that its can be also related to size and surface area of the fillers. We’ve added the description in Page 7, line 233~240.
Comment 3:
line 203 – additional space
Response: Thank you for pointing this fault. We have corrected it.
All of the changes were marked red in the manuscript.
We appreciate for your warm work earnestly, and hope that the correction will meet with approval. Once again, thank you very much for your efficient work and your constructive comments.
Best wishes.
Sincerely yours,
Fang Wang, Wenbo Shi, Yuliang Mai, Wei Hu and Bing Liao

Round 2
Reviewer 2 Report
The article has been redrafted and solidly improved. I appreciate the authors' contribution to the changes that have been made. In my opinion, the article is now ready for publication.